# Partnership between Chinese Dance Sport Couples: A Consensual Qualitative Research Analysis

**DOI:** 10.3390/ijerph192215369

**Published:** 2022-11-21

**Authors:** Xiuxia Liu, Guan Yang, Shen Wang, Xiangfei Wang

**Affiliations:** 1Post-Doctoral Research Station, Wuhan Sport University, Wuhan 430079, China; 2School of Physical Education, South China University of Technology, Guangzhou 510641, China; 3School of Physical Education and Sport Science, Fujian Normal University, Fuzhou 350117, China; 4Institute of Sport Science, Wuhan Sport University, Wuhan 430079, China

**Keywords:** partnership, dance sport, athlete, consensual qualitative research

## Abstract

The aim of our study was to explore the conceptions related to partnership between dance sport couples. We conducted in-depth interviews with 20 registered athletes of the Chinese Dance Sports Federation (CDSF) about partnership between dance couples, using the Consensual Qualitative Research (CQR) method. Results revealed that partnership tended to stem from seven domains: (1) mutual understanding, (2) instant intimacy, (3) long-term affection, (4) obligational ties, (5) instrumental ties, (6) tacit factors, and (7) mutual self-disclosure. Each domain included several categories of core ideas, most of which were general and typical across the respondents. The general and most typical core ideas were related to mutual understanding in all aspects (17T), pleasure (18T), sense of substitution (15T), intimacy (20G), harmony (20G), mutual help (20G), mutual tolerance (20G), mutual attraction (15T), responsibility (20G), training plan (12T), consistent goals (20G), skills improvement (20G), image matching (16T), mutual self-disclosure (18T), which suggested a general belief in the equity perspective regarding partnership between Chinese dance sport couples. Future studies need to examine diverse samples of athlete–athlete dyads to advance interpersonal theory in sports and add to emerging theories of performance behavior and expertise in sport.

## 1. Introduction

Starting out as one of the social graces expected of the European upper classes, “dance sport” (also called ballroom dance/competitive dancing), deriving from the Latin balare (to dance), is now an umbrella term for a style of lead and follow based snobbish and elite partner dancing [1]. As an extremely complex sports discipline [2], dance sport contains ten kinds of dances, namely Cha Cha, Samba, Rumba, Paso Doble, Jive, Waltz, Tango, Viennese Waltz, Foxtrot, and Quickstep. In addition, dance sport is danced both competitively and socially [3], from where there is a phenomenon of “feeling your partner” [4].

Dance sport requires the partners to follow the rhythm of the music and compete with multiple pairs of contestants to show the beauty of sports [5], so the overwhelmingly dominant aesthetic of partner dancing (and especially ballroom) is the pairing cooperation of male and female [3,6]. For the dancers, besides the great amount of practice (diligence and endurance) and a high level of technical expertise (for which a good trainer is important), the partner is vital. Good cooperation with that partner is of decisive importance for effective competition [7], which is supported by scholars from many countries [6,7]. Chinese scholars even argue that partnership is an essential characteristic of sports dance that distinguishes it from other sports [2,3]. Therefore, the amount of hours of ballroom coaching go into trying to depict the performing “male/female relationship” between partners [3], which is supported by the dyadic partnership between them. However, at present, few studies have explored the conceptions of partnership based on the characteristics of dance sport and cultural characteristics, which is happening in the context that athlete–athlete partnerships in dyadic sports, has been overlooked [8].

The current research explores conceptions of partnership between Chinese dance sport couples. Athlete–athlete partnership is a form of athletic dyad in which both members equally share power and responsibility [8,9]. The expressive ties and obligation ties between dance couples differentiate them from partnership in other dyadic sports. As a discipline based on romantic fantasy, dance sport expresses emotions openly [10], shows the body interaction between dance partners, and highlights the intimate relationship between men and women [11].

### 1.1. The Concept of Dance Sport Partnership

The partnership between a dance sport couple is hard to define, for the reason that it is a very close relationship, both on the physical level, as mentioned earlier, and on a personal level, having the harmonious characteristics typical of a work environment. The areas of professional interaction and personal life between dance partners are highly integrated—dance partners define each other as best friends, many of them are even life partners. Even so, some scholars try hard to explore the concept of partnership between dancesport couples.

Search terms included “competitive dancing” or “sports dance/dancing” or “sport dance/dancing” or “dance sport/dancing” or “latin dance” or “American latin dance/dancing” or “standard dance/dangcing” or “ballroom dance/dancing” or “athletics performance” or “partner relationship” or “dance sport partnership” or “dance sport and couple” or “dance sport and dyad” or “coach-athete relationship” or “intimate relationship” or “cпopтивнo-бaльныx тaнцax (ballroom dancing)” or “El Baile Deportivo (dance sport)” or “partnership” et al. As the theme word, we searched for literature in Google scholar, EBSCO, Web of science, Proquest, Riss and other databases, finding researchers in just 16 countries (Spain, India, South Korea, Hungary, the United States, Canada, the United Kingdom, Romania, France, Germany, Slovenia, Switzerland, Lithuania, Poland, Israel, and China) have conducted researches on dance sport partnership. Among them, only Korean scholars extracted the dimension of partnership and developed corresponding evaluation tools [12,13].

Based on grounded theory and a survey of 389 Korean sport dancers, Myung, Je-Min․ Kwak, Sung-Hee, and Seong Chang-Hoon first noted that partnership between dance sport couples is “a bilateral relationship based on mutual trust, shared benefits and risks between athletes, and equal responsibility for performance”, and distilled its five dimensions, namely partner care, partner harmony, reciprocal endeavor, perfect rhythm, and economic environment. Partner care means being patient and considerate with your partner and trying not to feel uncomfortable when you are together. Partner harmony means understanding your partner’s personality, communicating well, and connecting to your partner’s heart. A reciprocal endeavor requires a cooperative attitude from the partner, cooperating with frequent participation in training. Perfect rhythm means having a strong sense of music and technical tension. The economic environment refers to having the financial means to pay for training, classes, and competitions. In general, the achievement of sports dance is based on inviting teachers for long hours of classes and expensive clothes [12]. Based on this groundwork, Kim, Eun-Sug et al. proposed that partnership between dance sport couples is a collaborative relationship that can produce beautiful movements and sincere expression of their emotions. Characterized by mutual understanding and cooperative spirit, partnership includes four dimensions: trust, referee–audience feedback, physical harmony, and economic conditions. Trust is the basic requirement of partnership. Referee–audience feedback is a must for dance partner performance. Physical harmony requires the dance partner duo to match each other in appearance, physique, body proportions, body shape, and lines. The meaning of economic conditions is consistent with that mentioned by Myung, Je-Min et al., 2010 [13]. Based on the four dimensions mentioned above, Kim, Eun-Sug et al. developed a partnership scale for Korean competitors with a total of 18 items using 323 sport dancers registered with the Korean Sports Dance Association [13], and this scale has been utilized by other studies [5,14].

However, the dimensions mentioned above still have three limitations. 

(1) Some of the existing dimensions do not belong to the category of interpersonal relationships, since they can hardly reflect the psychological relationship between partners, they are difficult to be regarded as a component of partnership. For example, the economic environment and perfect rhythm (which are influencing factors for the formation and maintenance of the partnership) and referee–audience feedback (which is a factor that is affected by the competitive performance of the pair dancers). 

(2) Neglecting the characteristics of dance sport and studying the dimensions of partner relationship, such as instant intimacy. As Patrick Gaudreau et al. [15] mentioned, many dynamics of the dual partnership are unique to the sports culture they are engaged in; dance sport is a subject based on romantic fantasy. It openly expresses emotions [10], shows the body interaction of dance partners, and highlights the intimate relationship between men and women [11]. Compared with other paired sports such as table tennis mixed doubles, tennis mixed doubles, and figure diving, the cooperation between dance sport couples is very different [5,12]. 

(3) Few studies have paid attention to the cultural characteristics of dance sport partnership from the perspective of cultural psychology, which indicates that interpersonal relationships have cultural characteristics, as does partnership between dance sport couples. For example, based on the Rasch model in item response theory, Yang, Eun-Sim found that the statements like “my partner and I will be full of vitality when dancing in front of the crowd” and “my partner and I have left a deep impression on the audience” [13] were considered to be prejudiced against the male group for South Korean men rather than the object of appreciation [16]. As for the Chinese context, the obligations ruled by “reqing” which focuses on the partnership itself, without always considering instrumental intentions, has not been recognized.

### 1.2. The Present Study 

Although the study of athlete–athlete partnership frameworks is a very important research field [9,17], athlete–athlete partnership research is still in its infancy, especially between Chinese dance sport couples. The most important work of researchers remains in the creation of a comprehensive and ambitious theoretical system, where attention is often focused on the conceptual construction of the partnership [8]; therefore, it is urgent to explore its dimensions and components. In addition, belittling or ignoring the partnership between sport couples might also have a negative impact on performance (e.g., persistence, motivation, success) and mental health (e.g., happiness, satisfaction, anxiety) [18]. So, the purpose of our study is to conduct in-depth interviews with 20 outstanding Chinese sports dancers through consensus qualitative research (CQR) to explore the connotation and conception of the partnership between Chinese dance sport couples.

## 2. Materials and Methods

This study adopted the Consensus Quality Research (CQR) method [19,20], which absorbed the elements of phenomenology [21], grounded theory [22], and comprehensive process analysis [23]; this method helped us to achieve the goal of concept construction by finding the experience commonality among participants. However, CQR is different from other qualitative studies: (1) it requires an analysis team to participate in the discussion and reach a consensus on the data analysis results. On the one hand, various viewpoints and experiences among team members might help to solve the complexity and ambiguity of data. On the other hand, the use of consensus technology could improve the quality of decision-making [24,25]. (2) Try to keep the qualitative analysis method “loyal to the text”, to mine information from the original data, and to avoid the cognitive bias of the analyst’s “subjective assumption” [19]. (3) There is a relatively clear operation process, and on the basis of carefully defining the subjects, the unified program is used to collect data at one time. (4) Combined with frequency statistics on the basis of classification, the density or importance of each classification can be presented intuitively. (5) Pay more attention to the classification of subjective experience and behavioral responses, and pay less attention to the relationship between classifications [26].

Considering the fact that the first author of this paper has been involved in dance sports training and academic research for 7 and 5 years, respectively, our study was prone to interference from her personal experience, which made the study violate the norms of qualitative research and caused the reliability and validity of the results to be greatly reduced. Therefore, we chose the consensus analysis method to conduct the study, and the specific CQR implementation process is described below.

### 2.1. Participants

The sample for our study consisted of 20 registered athletes of the Chinese Dance Sports Federation (CDSF) (see Table 1 for demographic characteristics). Participants provided signed informed consent forms, and all agreed to be involved in the study. The sampling principles were as follows: (1) all participants had very intensive and rich information on dance sport partnership. As Hill et al. [19,20] suggested when selecting samples, researchers should randomly select a sample of 8 to 15 participants with rich knowledge of the research issues, so, we adopted the specific sampling strategy under the “purposive sampling” principle in qualitative research for this study. Finally, we chose the participants who had participated in training for more than 7 years, had spent at least 3 years with their partners, and had obtained the top 6 results in professional, professional-sequence, A-sequence, or other excellent groups. In our interviews, these participants showed a deep understanding of the dance sport partnership. (2) Sampling while analyzing the interview contents and stopping sampling when no new concepts were generated subsequently. In addition, we took other participants for the interview and verified that the interview data did not distill new concepts [22]. Based on the sampling principle, firstly, we stopped sampling after the 14th participant was interviewed. Secondly, 6 participants were selected for interview, and the content of their text materials was analyzed. It was found that the new concept could not be extracted, so we ended the interview. In addition, we invited Tian Tian, a postgraduate student from the Institute of Psychology of the Chinese Academy of Sciences (CAS) with 20 years training and experience in dance sport, and also the champion of Latin dance in the Chinese professional group for three consecutive years from 2013 to 2016, to evaluate whether the theories analyzed in this study had reached saturation; her answer was yes. 

The total interview time for these 20 athletes was 796 min, covering 45 days from 29 May 2019 to 12 July 2019. The reason for the long span was that most of the athletes participated in many competitions, so it was difficult to make an appointment. In particular, when athletes competed in foreign countries, jetlag would be considered, which set another obstacle for the interviews.

### 2.2. Researchers

The consensual qualitative research team was composed of five people: one female postgraduate student from the Institute of Psychology of the Chinese Academy of Sciences (CAS), who was the champion of Latin dance in the Chinese professional group for three consecutive years from 2013 to 2016. She even broke the record of Chinese athletes in international competitions and danced for 20 years; one male graduate student and two postgraduate students (one male, one female) in psychology from the Beijing Normal University, and one male doctor of education. Among them, the female postgraduate student from the Institute of Psychology of the Chinese Academy of Sciences developed the investigation protocol, oversaw data collection, and reviewed results at all stages of coding and final audit. The male graduate student led the data analysis, which involved the coding of the domains and core ideas and the cross-analysis. The two postgraduate students coded the domains and core ideas independently. The male doctor conducted both the initial audit of the codes and the final audit of the cross-analysis.

However, given that the champion of Latin dance had familiarity with the relevant research literature and may have potentially overly-subjective ideas about partnership in dance sport, she was not directly involved in the very early phases of the CQR, where the initial codes were derived from the cases. In this regard, the consensual qualitative research team discussed their anticipated outcomes prior to the first round of data analysis, but they suspended their preconceptions during all the phases of the analysis. The postgraduate students who had no knowledge of the research literature or dance sport training experience derived the themes and categories based on what they read from the data, but were also encouraged to modify and improve the categories during the coding and consensus process. Proposed changes were discussed, and the team members agreed on all codes, categories, and changes to these before they were finalized.

### 2.3. Procedure

#### 2.3.1. Previous Work

Firstly, we obtained the consent of the interviewees and explained the purpose. To be more specific, as the first author of this research was engaged in the teaching and training of dance sport for many years, it was easy to get in touch with the sport dancers. They were telephoned and explained the purpose and content of the study and the confidentiality of their information. After obtaining their consent, we arranged the interview time and place with the interviewees.

Secondly, we distributed the interview outline to the interviewees 3–5 days in advance. The participants were asked to answer 6 open-ended questions: (1) What do you think of the ideal partner? (2) Why did you become partners? (3) Were there any special events that affected your relationship? (4) How did you communicate or interact during training and competition? (5) What information will you share with each other? (6) Was there other important information about the partner/partner relationship?

Thirdly, we prepared interview tools, including a tape recorder, a paper version of interview outline, and a pen.

#### 2.3.2. Data Collecting

We collected data as the steps below: (1) we confirmed that the recorder was running normally, and the interview outline and pen were brought; (2) we ensured that the interview content was presented anonymously and only for academic use; (3) we showed respect for the opinions of the participates during the interview. If they felt uncomfortable about the questions, they could pause or refuse to answer questions at any time. (4) After permission, the talks were recorded; (5) we aimed to be sincere, open, and strived to create a more comfortable chat atmosphere; (6) we asked questions according to the interview outline and aimed to be be good at questioning to make sure that the participants could give effective information and answer questions pertinently. If there were difficulties in the answer process, for example, the participants could not understand the meaning of the question, they could be guided and assisted by reasonable questioning; (7) we interviewed with an objective and neutral attitude when guiding and assisting the participants to answer questions, they should be allowed to express their personal opinions and not comment on the spot, so as to respect the real psychological state of them; (8) we tried to use paper and pen to write down the key words and sentences of the conversation without permission to record. Finally, the interview materials were transcribed and sorted out. All the interview text materials of 20 interviewees totaled about 115,869 words, as shown in Table 1.

### 2.4. Data Analysis

According to the consensus quality research (CQR) method, the data analysis proceeded as follows: coding of domains, coding of core ideas, initial audit, cross-analysis, final audit, and stability check. In all these phases of data analysis, it was vital that the coding team arrive at a consensus regarding the classification and meaning of the data. 

Domains were topics used to group, cluster, or segment the data [20]. They represented the general themes commonly mentioned by participants in all items of the questionnaire. We started with an initial list of domains using 17 cases. The participants’ responses were classified under these initial domains. Each domain was represented by a letter (for example, “A” represented an initial domain called “factors that lead to tacit between the dance partners”). As more cases were analyzed, relevant research team members continued to examine and modify these areas to fit the data. Some domains were combined, separated into multiple domains, or created depending on the new information that was added with each case or participant.

Subsequently, we built core ideas for each domain. Core ideas are summaries of the data that capture the essence of the responses [20]. The purpose of this step is to extract each case or participant’s answers to its core components so that their responses were more comparable to other cases. Each core idea was represented by a number (e.g., 1 represented the core idea “Tacit understanding of actions”). In our case, one short answer from a single item sometimes contained more than one core idea. Each of the participants’ responses was then coded to classify them to a particular domain and core idea (for example, the respondent’s answer to item 2 was coded A1, which means that it was a factor that led to a tacit understanding between partners and specifically discusses the core idea of tacit understanding of movement).

Once these initial codes were set up and derived from the initial 17 cases, the two coders used these codes in the remaining cases. Each coder independently coded a specified number of cases, and then the two met to discuss and form a consensus on their codes. The interim audit was performed by a study team member who was not involved in the initial coding. This process ensured that the domains were adequate and that the core ideas represented the responses. When the auditor disagreed with the domain labeling and coding, discussions were held with other study team members until a consensus was reached.

The next step was cross-analysis. In this stage, the core ideas coded under the particular domains described above were analyzed into higher-level categories, which were extracted from the core ideas of all cases by determining which core ideas were similar or represented similar meanings. The cross-analysis was conducted independently by two research team members, who then met and discussed until they reached a consensus on all category labels.

We counted the frequency of respondents who mentioned specific categories when the category labels were agreed upon. Through this step, the team was able to determine which categories were more frequently mentioned by respondents. Hill et al. [20] suggest using the frequency labels “general”, “typical”, “variant”, or “rare” to characterize the data. “General” was applied to ideas mentioned by 19–20 of the 20 participants. “Typical” was applied to ideas mentioned by 11–18. “Variant” was applied to ideas mentioned by 4–10. “Rare” was applied to ideas mentioned by 2–3, and if it was applied to ideas mentioned by 1, the category was placed in the notes and was not reported in the results. 

Next to the final audit, the auditor provided feedback on the initial round of cross-analysis. Codes, category labels, frequency counts, and consensus judgments were examined. In cases of disagreement, the research team returned back to the initial responses and discussed whether the categories represented core ideas. The approach to data coding and category creation was iterative. 

The final step was to make stability checks. Seven responses were retained during the initial coding and cross-analysis of the domains. Responses from these cases were coded to determine if these categories covered all the data in our sample. No significant changes in categories were noted, therefore, we agreed that the results were stable.

## 3. Results

Data analysis formed six domains: (1) mutural understanding; (2) instant intimacy; (3) long-term affection; (4) obligational ties; (5) instrumental ties; (6) tacit factors; and (7) mutual self-disclosure. See Table 2, Table 3, Table 4, Table 5, Table 6 and Table 7 for a complete list of domains, categories, and frequencies.

### 3.1. Mutual Understanding

Mutual understanding refers to a kind of behavior and state that dyadic dancers consciously understand each other. It will help to build a good connection between partners. Without mutual understanding, it will be hard to promote a favorable partnership. This domain was divided into five categories: mutual understanding in all aspects (17T), in dance philosophy (15T), in habits and disposition (7V), in dance flaws (5V), and in disposition (2R) (see Table 2). 

**Table 2 ijerph-19-15369-t002:** Domains and categories.

Domains/Category	Core Ideas	Frequency
Mutual Understanding		
All aspects	We need to understand each other in all aspects so as to facilitate communication.	17T
Dance philosophy	Our dance ideas and ideas should be consistent.	15T
Habits and disposition	He knew I didn’t like parties, so he just turned me down.	7V
Dance flaws	After the competition, we will immediately summarize the problems and find ways to make up for each other, instead of each other’s problems.	5V

Note: categorized into T (typical) means that 11–18 participants mentioned this core idea; V (variant) means that 4–10 participants mentioned it.

Mutual understanding in all aspects was the most commonly mentioned item in this domain (*n* = 17). One participant stressed that no matter whether in life or in dance, more communication was needed to avoid conflicts caused by misunderstanding; it was easier to communicate by knowing your partner in all aspects. This may be due to the high combination of the professional sphere of interaction and personal life between dance sport partners. Partners defined each other as best friends, and often they were also life partners. Therefore, knowing each other in an all-round way, the partners can better predict the other party’s psychological behavior tendency and adjust their own behavior accordingly. Mutual understanding in dance philosophy was stressed by three-quarters of the participants. Only when the partners achieve the same style and state in the dance can a sense of discord be avoided. One participant emphasized that no matter what life was like, he must understand how his partner feels about the dance. Mutual understanding of habits and disposition was mentioned by nearly half of the participants. For example, they said, “they had known each other for many years, and they were familiar with each other’s habits in their life, so their partnership was maintained well”, or “it is very important to know what kind of person the partners are”. This may also be due to the demand for partnership generated by multiple role concentration. In addition, mutual understanding of dance flaws was stressed by 1/4 of the participants; some participants mentioned that they envied those dance partners who communicated with each other about their dance problems after the competition because, in this way, partners could know in time what was wrong with their cooperation and then they can adjust themselves over time, so as to contribute to the improvement of competitive performance; otherwise, sport dancers would make slower progress. Actually, “practice by competition” was encouraged. According to the competition requirements, the sport dancers must participate in several sub-races to obtain the total score and the qualification for the final competition. It was a necessary training method to improve technical movements and competitive performance through the competition. If the dance problems can not be found out in the competition and solved, then the value of the competition will be greatly reduced.

### 3.2. Instant Intimacy

Instant intimacy refers to a short-time passionate state of desire formed between dance partners in the competition context, which is characterized as pleasant pleasure under a state of desire for their partners. This domain was divided into three categories: pleasure (18T), sense of substitution (15T), and feeling of freshness (7V) (see Table 3). 

**Table 3 ijerph-19-15369-t003:** Domains and categories.

Domains/Category	Core Ideas	Frequency
Instant Intimacy		
Pleasure	I was very attentive and happy when I practiced dancing with her. The technical cooperation was good.	18T
Sense of substitution	If the relationship between the dance partners is particularly good, I will be very excited and eager to practice, so the effect of dance expression will be very good.	15T
Sense of freshness	She/he gives me a feeling that she/he can not feel dancing with me anymore.	7V

Note: categorized into T (typical) means that 11–18 participants mentioned this core idea; V (variant) means that 4–10 participants mentioned it.

Pleasure was mentioned by 18 participants in this domain. The pleasure refers to a psychological state of physical and mental relaxation when partners get along. Participants stressed, “We are the best friends, the best friend of the opposite sex. Because he is not irritable, we get along very easily, and dance with him will be very comfortable”, or “we cooperated well in technical when I was in a good mood and happy”. However, the decreasing sense of pleasure would always lead to a negative effect on the partnership and the results of the competition. For example, the participants mentioned, “if the partners did not adjust their moods, it would have a great negative impact on their performance”. Or “when she danced in a bad mood, I would be very tired and could not enter the training state”. A sense of substitution was stressed by 15 participants in this domain. The sense of substitution required the dancers to substitute the emotional needs given by the role into the cooperation between dance partners, which helped to improve the artistic expression. This was consistent with Yang, Eun-Sim’s view that passion and immersion were needed between dance sport partners [14]. For example, the participants believed that in order to perform well, the dance state and emotion must be put into the role, and only when two dancers’s hearts were together could they show better works to the audience. But, if they just danced their own double skill, they could not reach a higher artistic level. From the perspective of the mirror neurophysiological mechanism, the sense of substitution was the response of body language in sports dance. The reason was that the inferior parietal lobule of the human cerebral cortex, the broca region before ventral motor, and the posterior part of the inferior frontal gyrus has a “mapping” function of mirror neurons [27,28]. This mapping function transformed the closely connected crotch, closely attracted eyes and breath, as well as passionate feelings into an emotional booster between the partners. In the elegant passion, Yale University teacher, Sally Peters almost exaggerated the behavior of sport dancers, “driven by passion, their life revolves around dancing; and with enthusiasm and dedication, they practice complex patterns again and again. Whether on the dance floor or off the dance floor, they are consumed by their own imagination of dancing, such as practicing courtship rituals, being in love in tango, so that people can see attractive and attractive relationships at a glance. Outside the ballroom, are they still partners?” [29]. The feeling of freshness was mentioned by seven participants in this domain. It was a feeling that the dancers had for their partners, which would stimulate their desires to explore and activate the vitality. One male participant even mentioned, “She is different from others, I always feel something that others cannot give me, which drives me to collide with her in the dance”, and some participants stressed that just like the husband and wife described by others, they were lacking in some freshness when dancing together.

### 3.3. Long-Term Affection

Long-term affection refers to emotional ties generated in long-time interactions in life and professional contexts between dance partners, which is different from instant intimacy for its weaker emotional concentration and slower emotional outburst. This domain was divided into seven categories: intimacy (20G), harmony (20G), mutual help (20G), mutual tolerance (20G), mutual attraction (15T), mimetic relatives (10V), and close family ties (2R) (see Table 4).

**Table 4 ijerph-19-15369-t004:** Domains and categories.

Domains/Category	Core Ideas	Frequency
Long-Term Affection		
Intimate	Close emotions affect our expressiveness/I want to develop a close boyfriend/girlfriend relationship with her.	20G
Harmony	Harmony directly affects our training and competition performance.	20G
Mutual help	She ignored her own ideas and accompanied me to complete my heroic dream. I must take the crown on her head. If I had to change to a better partner, I would rather lose this aura and retire.	20G
Mutual tolerance	I stepped on the wrong music on the court and apologized to him. He didn’t blame me. He would patiently teach me after the game, so that next time I wouldn’t be wrong in the same place.	20G
Mutual attraction	Appreciation between two people will trigger different dancing feelings.	15T
Mimetic relatives	We have been together for a long time and played together. He is like my family.	10V
Close family ties	I have a close family relationship with my dancing partner. Our parents usually communicate and watch our competitions, and care about us very much. So we’ve always had a good relationship	2R

Note: categorized into G (general) means that 19–20 participants mentioned this core idea; T (typical) means that 11–18 participants mentioned it; V (variant) means that 4–10 participants mentioned it; R (rare) means that 2–3 participants mentioned it.

Intimacy was mentioned by 20 participants in this domain. It refers to a sense of close relationship formed by the accumulated training and life interaction between dance partners. Participants mentioned that “Many people may dance together because they have affections”, or “we were childhood sweethearts and danced together since childhood”, or “the experience of going out for training and competition would deepen the partnership, especially in foreign countries. Because humans were emotional animals, when one witnessed something and experienced something with their partner together, their partnership became closer and better”. 

Harmony refers to the coordination and accommodation between dance partners in terms of character and living habits, which was stressed in this domain. Participants stressed that harmony directly affected the quality of dance performance, and unharmonious living habits made it hard for them to go far, let alone achieve their common goals, and their character was important as well. Mutual help represents highly viscous ties characterized by dedication between partners. One participant stressed that they spent the first five or six years of dancing together in Shanghai. During these days, they helped each other a lot. Some other participants mentioned that they came and went together all the time—whenever they left home for school or came back home from school—they were the closest partners besides their family members, and they will take care of each other for life. Mutual tolerance aims to eliminate personality differences and ensure smooth communication. Participants stressed that mutual tolerance and concession were linking them tightly like a tie. Both dancers must understand their partners and be tolerant of each other, only in this way can they dance well. Otherwise, the cooperation will not last long. Mutual attraction refers to a strong psychological connection with appreciation between partners. Participants stressed that they hoped they got a more mature person. Some female participants stressed that they wanted to have a male partner who was more courageous and good at making decisions than themselves, daring to challenge them so that they would cultivate a sense of worship towards their male partners. In fact, this kind of worship plays an essential role in their training and performance. It would be hard to persist without mutual appreciation. In addition, mimetic relative and close family ties both reflected the closeness between dance partners from different aspects. Some participants mentioned, “I have a close family relationship with my dancing partner. Our parents usually communicate and watch our competitions, and care about us very much. So we’ve always had a good relationship”, or “he takes good care of me. Anyway, he is like a father in my heart”, or “partners are not only friends but also partners like family members, they can care about each other”. Even some female/male participants called their male/female partners as “good brothers”.

### 3.4. Obligational Ties

Obligational ties mean that dancers consciously abide by the etiquette with their partners. This domain was divided into four categories: responsibility (20G), training plan (12T), appreciation (10V), and respect (7V) (see Table 5).

**Table 5 ijerph-19-15369-t005:** Domains and categories.

Domains/Category	Core Ideas	Frequency
Obligational Ties		
Responsibility	If I don’t learn well, I will feel sorry for myself and him. Because he helped me technically, you failed to live up to his expectations.	20G
Training plan	Since 2014, we have really realized the importance of planning for a pair of dancers. We will make detailed plans, such as every overseas plan, domestic plan, training plan, and then follow them.	12T
Appreciation	I thank her very, very sincerely. Because I think I have a girl like this in my life who can accompany me from childhood to the present, standing at the top of this industry. I feel very happy and grateful.	10V
Respect	I also told him that I wanted to dance with my boyfriend after he entered college. He said that “I respected you and I would respect you as long as I told him in advance.	7V

Note: categorized into G (general) means that 19–20 participants mentioned this core idea; T (typical) means that 11–18 participants mentioned it; V (variant) means that 4–10 participants mentioned it.

Responsibility was mentioned by all participants in this domain. The sense of responsibility was the inherent requirement of social norms and role norms which is under the Chinese interpersonal relationship interpretation framework of “everyone is for me, I am for everyone”, such as matching with partners in terms of investment, ability, etc., and living up to the help of the dance partner. To coordinate with the social norms, some participants responded, “I’m a boy. I should take more care of her”, Or “the agreement between partners on dance must not be broken. Otherwise, it will greatly affect the partnership”, or “no matter who the partner is, if he appeared in the classroom to practice dancing and I don’t, I was guilty”, or “If I agreed to practice dancing, I would definitely go”, or “Dancing was something we must do in our life. Now when we retired from the position of national champion, we would also train ourselves and manage ourselves. Whether we were happy or tired, the base number of daily training is necessary”. The training plan requires the dancers to cooperate to formulate and follow the training plan, which was stressed by 12 participants in this domain. Cheng Dan, the champion of the Chinese professional group, stated, “Since 2014, we have really realized the importance of planning for a pair of dancers. We will make detailed plans, such as every overseas plan, domestic plan, training plan, and then follow them”. Appreciation was stated by 10 participants in this domain. Appreciation shows gratitude beyond the love expressed by partners’ companions, which is similar to the friendship between comrades in arms. For example, Cheng Dan pointed out, “I appreciated her very much from the bottom of my heart. She was such a girl in my life who can accompany me from my youth to the present, standing at the top of this area, I feel very lucky. Because of this, I always also tell myself to work harder”. Because the competition is full of uncertainty, dancers had to bear the risk of failure, the frustration of not reaching the top, and the regret of not making progress. In the whole process, dance partners were interdependent on each other, most of the dancers are filled with gratitude for their partner. Respect was stated by seven participants in this domain and it requires dancers to respect each other’s ideas in both life and training, and do not interfere with each other without permission. For example, the participants mentioned that they do not interfere in their partner’s personal affairs too much except for training, leaving some privacy. In addition, in sports dance training, the first author of this study found that dancers would fight in the training hall due to quarrels, leading to the termination of the dance partner relationship. Therefore, mutual respect is something that dancers should pay attention to.

### 3.5. Instrumental Ties

Instrumental ties mean that the dancers used the partnership as a means or tool to obtain achievements. This domain was divided into three categories: consistent goals (20G), skills improvement (20G), and image matching (16T) (see Table 6).

**Table 6 ijerph-19-15369-t006:** Domains and categories.

Domains/Category	Core Ideas	Frequency
Instrumental Ties		
Consistent goals	We spent a lot of money on foreign teaching for the competition, but less than half a month later, she told me that she was pregnant and had no way to continue the competition. At that time, my blood pressure was up. Later, we stopped dancing and broke up.	20G
Skills improvement	She has been to “Blackpool in England” and has seen a big scene. She can teach me technically.	20G
Image matching	I hope that the conditions of dance partners are as good as mine, which is conducive to the dance competition.	16T

Note: categorized into G (general) means that 19–20 participants mentioned this core idea; T (typical) means that 11–18 participants mentioned it.

Consistent goals were mentioned by all participants in this domain. The most important goal for sports dancers was that the dancers could cooperate with partners to improve their competitive ability and obtain excellent competitive performance. In addition, in order to achieve their goals, dancers are also required to match themselves in terms of external image. Some participants stressed, “Dancing partners were like partners in business”, or “we are mutually beneficial and win-win, when our goals are inconsistent, we will split up”. What is more, Tian Tian, the Latin dance champion of the Chinese professional group, pointed out that mature and excellent sports dancers do not have too high requirements for the high closeness of partnerships, but they paid more attention to whether there is a breakthrough in technology and achievements. Another participant said without hesitation, “I think the partner should go with me to get the results, otherwise don’t delay me for a second”. Therefore, inconsistent goals will also result in the breakdown of mutually beneficial and win-win relations. Only by developing each other’s talents can dancers gather them together and create synergy. Skills improvement was most commonly mentioned in this domain. One participant stressed, “The national standard dance is a kind of dyadic sport dancers would use their partners to improve their skills. To be frank, the partnership was just a kind of utilization relationship. You need her to continue dancing”. Image matching was most commonly mentioned in this domain. The dancers required that the external image of the partner should match with their own, and even most of the dancers could only accept better than their own. Participants stressed that since dancesport is artistic, a good-looking appearance is also conducive to their future development. When males chose females, they first required the partner to have good external conditions, and be almost acceptable technically.

### 3.6. Tacit Understanding

Tacit understanding refers to a high degree of harmony in the coordination of dance movements, emotions, and even overall performance. This domain was divided into three categories: dance movements (8V), emotional congruence (6V), and dance performance (5V) (see Table 7).

**Table 7 ijerph-19-15369-t007:** Domains and categories.

Domains/Category	Core Ideas	Frequency
Tacit Understanding Factors		
Dance movements	One look or movement can tell the other person’s meaning.	8V
Emotional congruence	When the partner is in a bad mood, we will consciously be quiet, and stop training to get rid of their bad mood, and continue to practice until the mood is getting good.	6V
Dance performance	He can catch my more creative “body thinking” and help me present my own ideas.	5V
Mutual Self-Disclosure	She had a hard time talking to her friends and family, so she just nagged me./A good performance requires a stable relationship with the dance partner, being sincere with the partner was the most important thing.	18T

Note: Categorized into T (typical) means that 11–18 participants mentioned this core idea; V (variant) means that 4–10 participants mentioned it.

Dance movements were most commonly stressed in this domain. Tacit understanding of dance movement can be generated in the accumulated cooperation training. Some participants stated that they had been working together for many years, and they know their partners’ next move from one look and one action. There are also young participants who were not satisfied with the degree of tacit understanding of their dance movements. They mentioned that sometimes when they cooperated with each other, they found that they were the same as their new partners, which made them very dissatisfied. According to the dynamic shaping theory of Pavlov, only when the movement cooperation between the partners reaches a certain degree of tacit understanding, then they will pay more attention to the overall artistic presentation in order to obtain better competition results. This requires the dancers to constantly cooperate with the practice in dance movements, until the motor-conditioned reflex system is consolidated and reaches the stage of establishing, consolidating the dynamic stereotype, and finally can achieve the unconscious production of accurate and beautiful movements. Emotional congruence was also mentioned in this domain; tacit understanding in emotional congruence was an accurate and effective perception of each other’s emotions formed in daily life and training, emotional congruence is the foreshadowing for the instant intimacy or long-term affection derived from the dance partnership. Participants stressed, “when she was in the back court of the game, I could tell if she was nervous at a glance. When she is nervous, I will be happy to amuse her”, or “she knew what I was doing without telling her”. Dance performance was mentioned by a quarter of the participants in this domain; tacit understanding in dance performance means that the contestants have realized the highest value attribute of dance-artistic value, which drove the dancers unconsciously looking for an impulse to release their most primitive feelings. One participant stressed, “When my partner dances with my emotions, we are like two train tracks suddenly connected. At this time, our emotions and actions will be released freely”. 

In summary, we believed that under the tacit understanding, the partners would output and receive information efficiently in the same training and competition, which is not only conducive to the maintenance and development of high-quality partnerships, but also to promote training efficiency and reach the optimal competitive performance.

### 3.7. Mutual Self-Disclosure

Mutual self-disclosure requires dancers to be sincere and show personal thoughts and feelings to their partners. This domain was also a category which means that partners can freely share their own thoughts, confusion, and occasional bad emotions and troubles with each other (see Table 7). For example, participants mentioned, “partner was a friend in life. When I was sick and sad, he would give spiritual and material care and support. As for us, we often participate in competitions far away from home or school, if he is not sincere to me, i will feel helpless”, or “basically we know about each other so well that we don’t need to put our thoughts in words. She knows everything about me. I know everything about her”, or “we will share various information with each other”, or “most of the time, dance athletes will continue to study abroad for three months and then continue to compete in China for three months”. For a long time, they can only company with each other, and almost can only share all their emotions with each other.

## 4. Discussion

The purpose of our study was to explore relationship components that contribute to the elite dance sport athlete–athlete partnerships. This exploratory study was one of few existing studies addressing the interpersonal components of athlete–athlete partnerships and one of two available studies examining partnerships between dance sport couples. Based on the analysis of the interview data of 20 excellent Chinese dance sport athletes by consensual qualitative research (CQR) method, it was found that partnership between dance sport couples in China includes seven constructs: mutual understanding, instant intimacy, long-term affection, obligational ties, instrumental ties, tacit factors, and mutual self-disclosure.

At present, the existing research has refined the components of partnership between dance sport couples, namely: partner care, partner harmony, reciprocal endeavor, perfect rhythm, economic environment, trust, referee–audience feedback, and physical harmony [12,13]. Among these components, instant intimacy was hardly mentioned. However, as a sport to express the intimate and passionate physical interaction and romantic relationship between men and women [10,11], instant intimacy is a very important component of partnership that encourages dancers to overcome difficulties and burnout, so as to continuously improve their competitive ability [30], and it was also related to whether dancers could have a better artistic performance in dance. In addition, instant intimacy was the requirement for partners under the mirror nerve mechanism, as mentioned in the results section of this study, “inferior parietal lobule of the human cerebral cortex, the broca region before the ventral motor, and the posterior part of the inferior frontal gyrus had the ‘mapping’ function of mirror neurons [27,28]. This ‘mapping’ function transformed the closely connected crotch, closely attracted eyes and breath, as well as passionate feelings into an emotional booster between the partners”. Therefore, sometimes dancers will be confused by the desire between the two sexes on the dance floor. Was it just a partner off the court? [29]. As a kind of athlete–athlete partnership that had different characteristics in different sports [8], instant intimacy was the essential feature that distinguished the partnership between dance sport couples from other sports. So it was never proposed in current studies.

As for obligational ties (responsibility, appreciation, training plan, respect) in partnership, its connotation included a reciprocal endeavor [12]. However, obligational ties also contained more meanings, such as responsibility and respect, which was different from the western one in the operating mechanism of psychology. To be more specific, obligational ties was ruled by Chinese “renqing” which relied on the moral power of society in the Chinese context after the partnership was established, dancers would respect each other, show responsibility, and abide by the training plan under the pressure of public opinion and social norms. If the voluntary contact was perceived, the partners would also express gratitude and train harder to obtain excellent competition results. Therefore, if one broke the rules, they felt ashamed. In this study, some participants even mentioned that “Once the partner arrives at the dance hall, he will feel guilty when he is not together”; however, obligations arise under the regulation of contracts in some western countries. Elite dancers in western countries would sign cooperation agreements with their partners.

The long-term affection (intimate, harmony, mutual help, mutual tolerance, mutual attraction, mimetic relatives, and close family ties) was in line with the partner care dimension of partnership between dance sport couples [12] and the closeness dimension in the 5C theory of athlete–athlete partnership [8], also in line with 3C theory of coach-athlete relationship [31] in terms of long-term intimacy. The reason was that as a “social person”, expressive ties were a gift given by society to everyone. For dancers, they often left their homes to go to other places for training. Without the support of any organizations, they had to help each other and tolerate each other, forming a harmonious relationship mode in the struggle process of sharing weal and woe, as well as forming close emotional ties, such as romantic relationships during this journey. When people are understood and appreciated, their intimacy will increase [32]. 

The instrumental ties (consistent goals, skills improvement, image matching) proposed in our study were determined by the characteristics of competitive sports, where competitive achievements were the key to power and status. On the contrary, athletes were forced to withdraw from the competition. In order not to be eliminated, they needed to be supported by their athletic achievements. Cheng Dan, the champion of China’s ballroom dance professional group, recalled: “At that time, our external image was not the most brilliant. We were not the most brilliant players, and no one knew our names. But we did not want to be eliminated, and instead of the best players, we were more active in this field of competition”. So it could be seen that dance sport partners were a community sharing common goals and tasks [33]. In this field of gold medal supremacy, partnership was often seen as a tool. This was consistent with the views of a Taiwan scholar [34], she stressed that partnership between dance sport couples was a kind of capital to complete training and competition. Through mutual cooperation, dancers achieved their utilitarian needs and proposed the type of instrumental partner. Highly interdependent partners emphasized the results of the exchange relationship of “gaining” and “giving” to each other, and the goals of both parties were utilitarian. So we believed that the utilitarian requirements of consistent goals, skills improvement, and image matching were vital elements for establishing, maintaining, and developing partnerships. 

Mutual understanding was proposed by our study, which was seldom presented by other research on partnership, we took it as an innovation point. We considered that individual communication was full of deviations and misreadings. In order to avoid frictions caused by cognitive deviation, dancers needed to accurately predict the internal psychology and behaviors on the basis of a full understanding of their partners, and make correct decisions for possible differences and conflicts. So Majoross believed that identifying and managing conflicts was critical and proposed the method, which included increasing communication and understanding between dance partners, giving tolerance, encouragement, and confidence to dance partners [7]. As for trust between dancers [13], we believed that it could be the concentrated embodiment of mutual self-disclosure. The latter could highly represent the meaning of the former. On the journey of mutual understanding, the dancers expressed their thoughts to each other. Under the effect of disclosure reciprocity, individual self-expression would trigger others to reveal their feelings [35]. Both mutual understanding and self-disclosure were designed to enhance the quality of partnership between dance sport couples. In addition, according to self-determination theory, the sense of relationship was an indispensable element that was born of people [36]. We also mentioned the other component of partnership between dance sport couples, i.e., tacit understanding, which was also seldom mentioned by relevant research. In addition, perfect rhythm, economic environment, and referee–audience feedback do not belong to the component of partnership; we do no discuss it too much.

The results of this study were suitable for Chinese excellent sport dancers who have more than 7 years of training experience, and have or have had 3 years of partner experience with their partners, and have obtained the top six results in professional, professional-sequence, A-sequence, and other excellent groups. However, we recommended further investigation on other sport dancers (those who have less than 7 years of training and no more than 3 years of experience with dance partners and have not obtained the top 6 results in professional, professional-sequence, A-sequence, and other excellent groups). Since the dancers in the sample reflected their experiences as young professional dancers or early international dancers, it was necessary to understand the internal components of dance partnerships in order to explore the interpersonal factors that have changed over time to promote advanced performance. 

There were, of course, limitations to the approach taken to meet the research aims in this paper. First, the consensual qualitative research (CQR) method adopted within this study had often been criticized as a “subjective assumption” [19], which made our research less reproducible. To mitigate this limitation, we try to keep the qualitative analysis method “loyal to the text”, mining information from the original data, and try to avoid the cognitive bias of the analysts, and strictly following the consensus requirements of the consensus quality research (CQR). 

Furthermore, a clear limitation of our study was that these concepts’ results cannot perfectly interpret the Chinese dance sport partnership. Although, we tried to tap the connotation of the partnership between Chinese sports dance couples and got some valuable findings. In an idealized form, the complex conceptual structure grasped by us with precise concepts existed in the changing minds of different individuals. Its components and meanings had extremely complex hierarchical differences among these individuals and contained countless differences and chaos full of sharp various conceptual relationships. Therefore, the highly abstract generalization of partnership between Chinese dance sport couples was just an idealized attempt, and the exploration of its’ contexts needed further exploration. Just like the 3C theory of the coach–athlete relationship [31], the 5C theory of the dyadic beach volleyball athlete–athlete partnership [8] and the 4-dimensional theory [13] and the 5-dimensional theory [12] of the existing partnership between dance sport couples, they were constantly enriched under the exploration of scholars. In addition, we suggested that practitioners and researchers interacting with this paper critically would consider our findings in their context and transfer them to other performance areas; In addition, they should consider whether similar findings will be confirmed if these methods are repeated in the athlete/worker population.

In addition, according to the opinion of sports psychologists, there were differences in the partnership between athletes and athletes in pairs events, which meant that when conducting research on partnerships in different events, we should carefully consider the risk of theoretical and practical incompatibility caused by transplanting the analytical framework of the partnership between couples in other events. It can be seen that we should strengthen the exploration of the internal dimension or analytical framework of the athlete–athlete partnership in different events. So, future studies need to continually examine diverse examples of athlete–athlete partnerships, so as to expand interpersonal relationship and performance theory in sport.

## 5. Applied Implications

Taylor, J., Ashford, M., Collins, D once stated that the importance of effective talent development in sport was well established as a key aspect of achieving high-level performance [37]. The connotation of partnership between Chinese dance sport couples in our study was obtained through retrospective interviews with competitive sports dancers. Its logical starting point was what kind of dance partnership promotes competitive performance. For example, the interview question—“what do you think of the ideal partner?” in our study is to collect information about the connotation of the dance partner relationship contributing to excellent competitive performance. Our study gives a new experience to changes and correction of the training process in dance sport. The use of research results can improve harmony and coordination between dance athletes. Some elements of this result can be implemented in the process on preparedness of the Chinese excellent sport dancers who have more than 7 years of training experience, and have or have had 3 years of partner experience with dance partners, and have obtained the top 6 results in professional, professional-sequence, A-sequence, and other excellent groups. 

In general, competitive dancers and dance sport coaches should realize the importance of the partnership between dance couples on competitive performance. Competitive dancers should take the initiative to promote competitive performance through the management of dance partner relationship. Coaches should pay attention to the change of dance partnership at any time while conducting technical teaching, and maximize the competitive performance of dancers by maintaining and improving the quality of dance partnership. To be more specific:(1)Create a romantic training atmosphere or a psychological connection between dance couples before the competition, if possible, thus enhancing the instant intimacy (pleasure, sense of substitution, feeling of freshness) between the partners. Even encourage the dancers to fall in love with their partners. The literature found that partners must exhibit passion and emotion during dancing; very often, high-class dancing couples are also pairs in life [7,38]. In addition, research on the social psychological attributes of dyadic participants found the performance of dyads was better if members liked each other [39].(2)Encourage mutual self-disclosure between the dance partners to promote mutual understanding and, thus, enhance their ability to perceive their partners’ psychological and behavioral tendencies so as to improve the quality of the training, even competitive performance. The reason was as follows: based on the analytical framework of the self-concept of “independent self” and “interdependent self” proposed by Marcus and Kitayama according to the characteristics of cultural differences, it is emphasized that easterners preferred to discover themselves by relying on others [40], which is an inclusive individualistic interpersonal relationship. Sampson emphasizes that the ego boundaries of such relationships are fluid, and that the establishment of expressive ties accelerates the blurring of such boundaries [41]. Chinese intimacy is the result of interaction between individuals after the dissolution of boundaries and the process of mutual aggression between them, followed by a state of “no distinction between you and me” [42]. So, the psychological connection between the Chinese dance sport partners is a combination of instrumental ties and expressive ties influenced by different laws of operation, where the instrumental ties occur in the training competition arena, the expressive ties occur in daily life, and the dancers are likely to turn the cooperation issue into a love issue, making the intimate partnership between the dyads in practice break the conventional mode of operation of professional cooperation. In the absence of contractual constraints, the maintenance and development of partnerships will be challenged [43].(3)For dancers who regard their partners as only a kind of training capital, remind them not to release this mentality too much to their partners, but to pay attention to the “renqing” and “mianzi” between them, showing the obligational ties (responsibility, appreciation, training plan, respect) with their partners, instead. The reason is that the obligational ties between couple cushions dancers from the lose and hollowness of being used as a tool. This is why Chinese utilitarian relationships are often attached by obligatory ties.(4)Cultivate the mutual tacit understanding in dance movements, emotional congruence, and dance performance. We suggested developing and offering relevant courses to improve the empathy and emotional intelligence of sports dancers in the future. As Scholars, based on Howard Gardner, a developmental psychologist at Harvard University, proposed that dancers’ interpersonal skills are the engine of artistic communication between dance partners [44].

In addition, it is also recommended to cultivate long-term affection between dance partners, such as intimacy, harmony, mutual help, mutual tolerance, mutual attraction.

## 6. Conclusions

Partnership between dance sport couples in China is a complex concept whose components are as the follows: mutual understanding (in all aspects, in dance philosophy, in habits and disposition, in dance flaws, in disposition), instant intimacy (pleasure, sense of substitution, feeling of freshness), long-term affection (intimacy, harmony, mutual help, mutual tolerance, mutual attraction, mimetic relatives, close family ties), obligational ties (responsibility, appreciation, training plan, respect), instrumental ties (consistent goals, skills improvement, image matching), tacit factors (dance movements, emotional congruence, dance performance), and mutual self-disclosure. Our findings extended research on athletic dyads, such as dance sport, in three ways: first, according to the project characteristics of sports dance expressing intimate physical and emotional interaction between male and female, we proposed the concept of instant intimacy, which was rarely mentioned in the previous athlete–athlete partnership. In addition, our research also emphasized the cultural characteristics of dance partnership, we explained that the obligation tie which was ruled by “renqing” was different from the obligation ties corresponding to the contract in the West. Second, the consensual qualitative research (CQR) method was first used in the research of athlete–athlete partnership, which provided a new research idea for later related research. Third, we should encourage athletes to explore ways to achieve outstanding performance by revealing the interpersonal elements in their interdependent development.

## Figures and Tables

**Table 1 ijerph-19-15369-t001:** Basic information of interviewed athletes.

Discpline	Code	Sex	Age (Year)	Triaining Time (Year)	Partner-Time (Year)	Domestic Best Results	Interview Time (Minute)	Interview Recordings (Words)
M	A	M	31	20	14	First (PRO)	65	11,935
M	B	F	25	12	6	Third (PRO)	45	6297
M	C	M	26	16	4	Forth (PRO)	43	4376
M	D	M	23	13	3	Forth (PRO)	39	1863
M	E	F	29	21	3	Sixth (PRO)	27	4693
M	F	F	25	18	6	Sixth (PRO)	34	4862
M	G	F	32	20	5	Second (PRO-S)	26	2717
M	H	M	26	20	7	Third (A-group)	67	9453
M	I	F	27	16	3	Third (A-group)	39	4018
M	J	M	20	15	8	Third (A-group)	36	4937
M	K	M	24	8	3	Forth (A(S)-group)	21	2332
L	L	M	28	10	7	Second (PRO)	50	8699
L	M	F	25	15	3	Third (PRO)	50	8500
L	N	F	22	10	3	First (A(S)-group)	53	11,012
L	O	F	21	10	8	Third (A-group)	26	3399
L	P	F	22	13	4	Eighth (A-group)	30	4219
L	Q	F	19	10	3	Sixth (A(S)-18 years-group)	32	3521
L	R	M	19	7	3	Sixth (A(S)-18 years-group)	43	8026
L	S	F	21	8	3	First (A(S)-group)	43	4468
L	T	F	21	15	3	First (A(S)-group)	27	6532

Note: M = modern dance; L = Latin dance; PRO = professional; PRO-S = professional sequence; A(S)-group = A-sequence.

## Data Availability

The data of the present study can be available from the corresponding author via reasonable request.

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
