# Peer review of "Partnership between Chinese Dance Sport Couples: A Consensual Qualitative Research Analysis"

_ijerph, 2022, doi:10.3390/ijerph192215369_

Round 1

Reviewer 1 Report

In my opinion the main task of the research is the ways of completing dance couples. The topic is original, because connection and partnership between dance couples is a problem which influences competition results. This is a study given a new experience to changes and correction of the training process on dance sport. The use of results of the research can improvement  harmony and coordination between dance athletes. Some elements of this results can be implemented in process o preperedness of elita dancers. I think that conclusions are consistent with the evidence. The references correspond to the results of the study.

I have three questions for the authors.

First, it seems inappropriate to compare dance sports and beach volleyball.

Secondly, the practical significance of the work carried out is not entirely clear. How research can be used in the practice of the training process in competitive dance?

Thirdly, the authors must clearly define the boundaries of the use of the results obtained.

Besides, Table 2 is very difficult to understand. It is desirable to simplify or split the table.

Author Response

Dear reviwer and editor :

On behalf of all the contributing authors, I would like to express our sincere appreciations of your letter and reviewers’ constructive comments concerning our article entitled “Partnership between Chinese DanceSport Couples:A Consensual Qualitative Research Analysis” (Manuscript No: ijerph-1999954). These comments are all valuable and helpful for improving our article. According to your comments, we have made extensive modifications to our manuscript. In this revised version, changes to our manuscript were all highlighted within the document by using red-colored text. Point-by-point responses to the nice reviewer are listed below this letter. We appreciate for your warm work earnestly and hope that the correction will meet with approval. In addition, we feel great thanks for your your nice suggestions, we have also made extensive corrections to our previous draft, the detailed corrections are listed below.

Comments: In my opinion the main task of the research is the ways of completing dance couples. The topic is original, because connection and partnership between dance couples is a problem which influences competition results. This is a study given a new experience to changes and correction of the training process on dance sport. The use of results of the research can improvement  harmony and coordination between dance athletes. Some elements of this results can be implemented in process o preperedness of elita dancers. I think that conclusions are consistent with the evidence. The references correspond to the results of the study.

I have three questions for the authors.

1: First, it seems inappropriate to compare dance sports and beach volleyball.

Response:

Thank you for your comments. We are very appratiated this advice, The corresponding revision is made in the revised version of manuscript.

Revision:

we deleted the sentences as following,“For example, as for beach volleyball and dancesport, It did not emphasize the aesthetic feeling of the movement between partners, and did not tend to present the aesthetic feeling of the interaction between two athletes. Dancesport was a skill oriented performance project. athleltes would try to cooperate with their partners in order to obtain the best competitive performance, and finally show the aesthetic feeling through artistic body movements. Therefore, the athlete- athlete partnership in beach volleyball mights not emphasize the passions between the sexes.”

2: Secondly, the practical significance of the work carried out is not entirely clear. How research can be used in the practice of the training process in competitive dance?

Response:

Thank you for your comments. The corresponding revision is made in the revised version of manuscript.

Revision:

On page 18-19, line 717-781 (revised manuscript), we added a section (5.Applied Implications) to stress the practical implications:

“5. Applied Implications

Taylor, J.; Ashford, M.; Collins, D once stated that the importance of effective talent development in sport was well established as a key aspect of achieving high-level performance [37]. The connotation of partnership between Chinese dance sport couples in our study was obtained through retrospective interviews with competitive sports dancers. Its logical starting point was what kind of dance partnership promotes competitive performance. For example, the interview question——“what do you think of the ideal partner? ” in our study is to collect information about the connotation of dance partner relationship contributing to excellent competitive performance. Our study given a new experience to changes and correction of the training process on dance sport. The use of research results can improve harmony and coordination between dance athletes. Some elements of this result can be implemented in process on preperedness of the Chinese excellent sportdancers who have more than 7 years of training experience, and have or have had 3 years of partner experience with dance partners, and have obtained the top 6 results in Professional, Professional-sequeny, A-sequeny and other excellent groups.

In general, competitive dancers and dance sport coaches should realize the importance of the partnership between dance couples on competitive performance. Competitive dancers should take the initiative to promote competitive performance through the management of dance partner relationship. Coaches should pay attention to the change of dance partnership at any time while conducting technical teaching, and maximize the competitive performance of dancers by maintaining and improving the quality of dance partnership. To be more specific:

(1) Create a romantic training atmosphere or a psychological connection between dance couples before the competition if possible, thus enhancing the instant intimacy (pleasure, sense of substitution, feeling of freshness) between the partners. Even encourage the dancers to fall in love with their partners. As literatures found that partners must exhibit passion and emotion during dancing. Very often, high class dancing couples are also pairs in life [7][38]. In addition, research on the social psychological attributes of dyadic participants found the performance of dyads was better if members liked each other [39].

(2) Encourage mutual self-disclosure between the dance partners to promote mutual understanding and thus enhance their ability to perceive their partners' psychological and behavioral tendencies, so as to improve the quality of the trainning, even competitive performance. The reason was as follows: based on the analytical framework of the self-concept of “independent self” and ”interdependent self” proposed by Marcus and Kitayama according to the characteristics of cultural differences, it is emphasized that easterners preferred to discover themselves by relying on others [40], which is an inclusive individualistic interpersonal relationship. Sampson emphasizes that the ego boundaries of such relationships are fluid, and that the establishment of expressive ties accelerates the blurring of such boundaries [41]. Chinese intimacy is the result of interaction between individuals after the dissolution of boundaries and the process of mutual aggression between them, followed by a state of “no distinction between you and me” [42]. So, the psychological connection between the Chinese dance sport partners is a combination of instrumental ties and expressive ties influenced by different laws of operation, where the instrumental ties occurs in the training competition arena and the expressive ties occurs in daily life, and the dancers are likely to turn the cooperation issue into a love issue, making the intimate partnership between the dyads in practice break the conventional mode of operation of professional cooperation. In the absence of contractual constraints, the maintenance and development of partnerships will be challenged [43].

(3) For dancers who regard their partners as only a kind of training capital, remind them not to release this mentality too much to their partners, but to pay attention to the "renqing" and "mianzi" between them, showing the obligational ties (responsibility, appreciation, training plan, respect) with their partners, instead. The reason is that the obligational ties between couple cushions dancers from the lose and hollowness of being used as a tool. This is why, Chinese utilitarian relationships are often attached by obligatory ties.

(4) Cultivate the mutual tacit understanding in dance movements, emotional congruence and dance performance. We suggested developing and offering relevant courses to improve empathy and emotional intelligence of sports dancers in the future. As Scholars, based on Howard Gardner, a developmental psychologist at Harvard University, proposed that dancers' interpersonal skills are the engine of artistic communication between dance partners [44].

In addition, It is also recommended to cultivate long-term affection between dance partners, such as intimacy, harmony, mutual help, mutual tolerance, mutual attraction.”

3: Thirdly, the authors must clearly define the boundaries of the use of the results obtained.Besides, Table 2 is very difficult to understand. It is desirable to simplify or split the table.

Response:

Thank you for your comments. we tried our best to define the boundaries of the use of our results. The corresponding revision is made in the revised version of manuscript.

Revision:

On page 17, line 672-675 (revised manuscript), we added the following contents: “The results of this study were suitable for the Chinese excellent sportdancers who have more than 7 years of training experience, and have or have had 3 years of partner experience with their partners, and have obtained the top 6 results in Professional, Pro-fessional-sequeny, A-sequeny and other excellent groups.”

In addition, we split Table 2 and disbuted them next to the relavant results, so that it can be seen more clearly. At the same time, we make some notes under the tables, so as to ilustrate the content of the tables, for example: “categorized into T(typical) means that this core ideas was metioned by 11–18 participants.”

We tried our best to improve the manuscript and made some changes marked in red in revised paper which will not influence the content and framework of the paper. We appreciate for Editors/Reviewers’ warm work earnestly, and hope the correction will meet with approval. Once again, thank you very much for your comments and suggestions.

Reviewer 2 Report

I would like to thank you for the opportunity to review the article submitted to the International Journal of Environmental Research and Public Health.

 I think that the manuscript is very poorly written. There are many errors throughout the text such as: missing spaces, incorrect quotation marks, inserted the dot instead of the comma. It is hard to read. In addition, style, statistical reporting, and reference citations should conform to the American Psychological Association’s guidelines, from the Publication Manual of the American Psychological Association (seventh Edition). Therefore, for example, it should be written (line 114): ‘As Patrick Gaudreau et al., [15] mentioned that many dynamics of the dual partnership are unique to the sports culture they are engaged in’.

Please refer to my specific comments below.
There are many such errors (mentioned above) throughout the text. For example:
Line 12: There should be a dot instead of a comma.
Line 43: The word - ‘relationship’ should be written in lowercase.
Line 47: The phrase - ‘at’ should also be written in lowercase.
Line 52: The phrase – ‘as’ should also be written in lowercase.
Line 71 and line 72: Chinese nomenclature (Chinese language) should be removed.
Line 84: You should delete the dot after the word ‘environment’. You should change the sentences.
Line 132-135: You should change the sentences. There are some punctuation errors. The same error is in lines 199 and 201.

In addition, please provide more information about the interview guide. Quasi-replication criteria should be provided.
Do you think that your sample utilising the notion of information power or the notion of data saturation? You should explain it in the text.

You should presents the power levels, using for example, the G*Power software supports sample size and power calculation for various statistical methods. Did not you calculate the minimum recommended size for your research? It is necessary to present the sampling error, the margin of error, as well as achieved confidence level of.

Another very important issue is that the limitations of the study and the practical implications should be formulated.

I recommend reading, for example, the article available at https://www.mdpi.com/2624-8611/4/4/50

I am pleased to wait for your revised version then.
Best regards.

Author Response

Dear reviewer and editor :

On behalf of all the contributing authors, I would like to express our sincere appreciations of your letter and reviewers’ constructive comments concerning our article entitled “Partnership between Chinese DanceSport Couples:A Consensual Qualitative Research Analysis” (Manuscript No: ijerph-1999954). These comments are all valuable and helpful for improving our article. According to your comments, we have made extensive modifications to our manuscript. In this revised version, changes to our manuscript were all highlighted within the document by using red-colored text. Point-by-point responses to the nice reviewer are listed below this letter. We appreciate for your warm work earnestly and hope that the correction will meet with approval.

1: I think that the manuscript is very poorly written. There are many errors throughout the text such as: missing spaces, incorrect quotation marks, inserted the dot instead of the comma. It is hard to read. In addition, style, statistical reporting, and reference citations should conform to the American Psychological Association’s guidelines, from the Publication Manual of the American Psychological Association (seventh Edition)2. Therefore, for example, it should be written (line 114): ‘As Patrick Gaudreau et al., [15] mentioned that many dynamics of the dual partnership are unique to the sports culture they are engaged in’.

Response :

Thank you for your comments. The English has been carefully polished and the revisions are highlighted in red We warmly invite respected you to review the revised articles again. And the format of reference [15] has been revised  ( line113-114 ).

2: Please refer to my specific comments below.There are many such errors (mentioned above) throughout the text. For example:
    Line 12: There should be a dot instead of a comma.

Response :

Thank you for your comments. It has been revised (see line 12).

Line 43: The word - ‘relationship’ should be written in lowercase.

Response :

Thank you for your comments.It has been revised (see line 44).

Line 47: The phrase - ‘at’ should also be written in lowercase.

Response :

Thank you for your comments. It has been revised ( see line 48).

Line 52: The phrase – ‘as’ should also be written in lowercase.

Response :

Thank you for your comments.It has been revised (see line ).

Line 71 and line 72: Chinese nomenclature (Chinese language) should be removed.

Response :

Thank you for your comments. It has been removed.

Line 84: You should delete the dot after the word ‘environment’. You should change the sentences.

Response :

Thank you for your comments.we changed the sentences. ( line 85).

Line 132-135: You should change the sentences. There are some punctuation errors. The same error is in lines 199 and 201.

Response:

Thank you for your comments. We changed the sentences and revised the punctuation errors. The corresponding revision is made in the revised version of manuscript.

Revision:

 On page 3, line 131-136 (revised manuscript), the sentence of “Although the study of athlete-athlete partnership frameworks is a very important research direction [9,17]. however, it was not until 2019 that Poczwardowski a, Lamphere B, Allen K et al established a 5C model of athlete-athlete partnership based on the characteristics of beach volleyball[8]. few scholars have explored athlete-athlete partnership, especially between Chinese dancesport couples. As it can be seen, athlete-athlete partnership research is still in its infancy. The most important work of researchers remains the creation of a complete and ambitious system in the theoretical wilderness, where attention is often focused on the conceptual construction of the partnership[8]. The same is true of the partnership between dancesport couples, so it is urgent to explore its dimensions and componets” has been revised to“Although the study of athlete-athlete partnership frameworks is a very important research field [9,17], athlete-athlete partnership research is still in its infancy, especially, between Chinese dance sport couples. The most important work of researchers remains in the creation of a comprehensive and ambitious theoretical system, where attention is often focused on the conceptual construction of the partnership [8]. So, it is urgent to explore its dimensions and components.”

3: Please provide more information about the interview guide. Quasi-replication criteria should be provided.

Response:

Thank you for your comments. we tried our best to provided more information about the interview guide and quasi-replication critera. The corresponding revision is made in the revised version of manuscript.

Revision:

On page 4 and page 5, line 166-193 (revised manuscript), we added the relevant explanations:

“The sample for our study consisted of 20 registered athletes of Chinese Dance Sports Federation (CDSF) (see Table 1 for demographic characteristics). They provided signed informed consent forms, and all agreed to participate in the study. The sampling prin-ciples were as following: (1) all participants had very intensive and rich information in dance sport partnerhip. As Hill et al., [19,20] suggested that when selecting samples, researchers should randomly select a sample of 8-15 participants with rich knowledge of the research issues. So we adopted the specific sampling strategy under the "purposive sampling" principle in qualitative research for this study. Finally, we chose the partici-pants who participated in training for more than 7 years, spent at least 3 years with their partners, and have obtained the top 6 results in Professional, Professional-sequeny, A-sequeny or other excellent groups. In our interviews, these participants showed a deep understanding of the dance sport partnership. (2) Sampling while analyzing the interview contents, stopping sampling when no new concepts were generated subse-quently. In addition, we took other participants for the interview, and verifying that the interview data did not distill new concepts [22]. Based on the sampling principle, firstly, we stopped sampling after the 14th participant was interviewed. Secondly, 6 partici-pants were selected for interview, and the content of their text materials was analyzed. It was found that the new concept could not be extracted, so we ended the interview. In addition, we invited Tian Tian, a postgraduate student from Institute of Psychology of Chinese Academy of Sciences (CAS) with 20 years training and experience in dance sport, also the champion of latin dance in the Chinese professional group for three consecutive years from 2013 to 2016, to evaluate whether the the theories analyzed in this study had reached saturation. Her answer was yes.

The total interview time for these 20 athletes was 796 minutes, covering 45 days from May 29, 2019 to July 12, 2019. The reason for the long span was that most of the athletes participated in many competitions. So it was difficult to make an appointment. In particular, when athletes competed in foreign countries, jetlag would be considered, which set another obstacle for the interview.”

4: Do you think that your sample utilising the notion of information power or the notion of data saturation? You should explain it in the text. You should presents the power levels, using for example, the G*Power software supports sample size and power calculation for various statistical methods. Did not you calculate the minimum recommended size for your research? It is necessary to present the sampling error, the margin of error, as well as achieved confidence level of.

Response:

Thank you for your comments. We are so sorry to make disdingushed you confused. We would like to explain that the Consensus Quality Research (CQR), which was developed by the Hill team, is a qualitative method, absorbing the elements of phenomenology, grounded theory and comprehensive process analysis (on page 4, line 143-158). The goal of theoretical construction by finding the experience commonality among participants. So we use the notion of data saturation. The process of sample selection was metioned on page 4 and page 5, line 166-193.

5: Another very important issue is that the limitations of the study and the practical implications should be formulated. I recommend reading, for example, the article available at https://www.mdpi.com/2624-8611/4/4/50

Response:

Thank you for your comments. According to the valuable opinions, we have studied your article, “The Role of Challenge in Talent Development: Understanding Impact in Response to Emotional Disturbance” very carefully, we find it very helpful to us to revise our paper. The corresponding revision is made in the revised version of manuscript.

Revision:

On page 18 line 718-719 (revised manuscript). we cited the idea “the importance of effective talent development in sport is well established as a key aspect of achieving high-level performance” from this paper as to emphasize that the premise of tapping the connotation of dance partner relationship is to promote competitive performance.

In addition,we added two sections to stress the limitations and the practical implications of our study.

On page 17, line 682-714 (revised manuscript), the limitations were added:

“There were of course limitations to the approach taken to meet the research aims in this paper. First, the Consensual Qualitative Research (CQR) method adopted within this study had often been criticised as “subjective assumption” [19], which made our research less reproducible. To mitigate this limitation, we try to keep the qualitative analysis method “loyal to the text”, mining information from the original data, and try to avoid the cognitive bias of the analyst's, and strictly following the consensus requirements of the Consensus Quality Research (CQR).

Furthermore, a clear limitation of our study was that these concepts results cannot perfectly interpret the Chinese dance sport partnership. Although, we tried to tap the connotation of the partnership between Chinese sports dance couples, and got some valuable findings. In an idealized form, the complex conceptual structure grasped by us with precise concepts existed in the changing minds of different individuals. Its com-ponents and meanings had extremely complex hierarchical differences among these in-dividuals and contained countless differences and chaos full of sharp various conceptual relationships. Therefore, the highly abstract generalization of partnership between Chinese dance sport couples was just an idealized attempt, and the exploration of its’ contexts needed further exploration. Just like the 3C theory of the coach-athlete rela-tionship [31], the 5C theory of the dyadic beach volleyball athlete-athlete partnershship [8] and the 4-dimensional theory [13] and the 5-dimensional theory [12] of the existing partnership between dance sport couples, they were constantly enriched under the ex-ploration of scholars. In addition, we suggested that practitioners and researchers in-teracting with this paper critically would consider our findings in their context and transfer them to other performance areas; In addition, they should consider whether similar findings will be confirmed if these methods are repeated in the athlete/worker population.

In addition, according to the opinion of sports psychologists, there were differences in the partnership between athletes and athletes in pairs events, which meant that when conducting research on partnerships in different events, we should carefully consider the risk of theoretical and practical incompatibility caused by transplanting the analytical framework of the partnership between couples in other events. It can be seen that we should strengthen the exploration of the internal dimension or analytical framework of the athlete-athlete partnership in different events. So future studies needed to continually examine diverse examples of athlete–athlete partnerships, so as to expand interpersonal relationship and performance theory in sport.”

On page 18-19, line 717-781 (revised manuscript), we added a section (5.Applied Implications) to stress the practical implications:

“5. Applied Implications

Taylor, J.; Ashford, M.; Collins, D once stated that the importance of effective talent development in sport was well established as a key aspect of achieving high-level performance [37]. The connotation of partnership between Chinese dance sport couples in our study was obtained through retrospective interviews with competitive sports dancers. Its logical starting point was what kind of dance partnership promotes competitive performance. For example, the interview question——“what do you think of the ideal partner? ” in our study is to collect information about the connotation of dance partner relationship contributing to excellent competitive performance. Our study given a new experience to changes and correction of the training process on dance sport. The use of research results can improve harmony and coordination between dance athletes. Some elements of this result can be implemented in process on preperedness of the Chinese excellent sportdancers who have more than 7 years of training experience, and have or have had 3 years of partner experience with dance partners, and have obtained the top 6 results in Professional, Professional-sequeny, A-sequeny and other excellent groups.

In general, competitive dancers and dance sport coaches should realize the importance of the partnership between dance couples on competitive performance. Competitive dancers should take the initiative to promote competitive performance through the management of dance partner relationship. Coaches should pay attention to the change of dance partnership at any time while conducting technical teaching, and maximize the competitive performance of dancers by maintaining and improving the quality of dance partnership. To be more specific:

(1) Create a romantic training atmosphere or a psychological connection between dance couples before the competition if possible, thus enhancing the instant intimacy (pleasure, sense of substitution, feeling of freshness) between the partners. Even encourage the dancers to fall in love with their partners. As literatures found that partners must exhibit passion and emotion during dancing. Very often, high class dancing couples are also pairs in life [7][38]. In addition, research on the social psychological attributes of dyadic participants found the performance of dyads was better if members liked each other [39].

(2) Encourage mutual self-disclosure between the dance partners to promote mutual understanding and thus enhance their ability to perceive their partners' psychological and behavioral tendencies, so as to improve the quality of the trainning, even competitive performance. The reason was as follows: based on the analytical framework of the self-concept of “independent self” and ”interdependent self” proposed by Marcus and Kitayama according to the characteristics of cultural differences, it is emphasized that easterners preferred to discover themselves by relying on others [40], which is an inclusive individualistic interpersonal relationship. Sampson emphasizes that the ego boundaries of such relationships are fluid, and that the establishment of expressive ties accelerates the blurring of such boundaries [41]. Chinese intimacy is the result of interaction between individuals after the dissolution of boundaries and the process of mutual aggression between them, followed by a state of “no distinction between you and me” [42]. So, the psychological connection between the Chinese dance sport partners is a combination of instrumental ties and expressive ties influenced by different laws of operation, where the instrumental ties occurs in the training competition arena and the expressive ties occurs in daily life, and the dancers are likely to turn the cooperation issue into a love issue, making the intimate partnership between the dyads in practice break the conventional mode of operation of professional cooperation. In the absence of contractual constraints, the maintenance and development of partnerships will be challenged [43].

(3) For dancers who regard their partners as only a kind of training capital, remind them not to release this mentality too much to their partners, but to pay attention to the "renqing" and "mianzi" between them, showing the obligational ties (responsibility, appreciation, training plan, respect) with their partners, instead. The reason is that the obligational ties between couple cushions dancers from the lose and hollowness of being used as a tool. This is why, Chinese utilitarian relationships are often attached by obligatory ties.

(4) Cultivate the mutual tacit understanding in dance movements, emotional congruence and dance performance. We suggested developing and offering relevant courses to improve empathy and emotional intelligence of sports dancers in the future. As Scholars, based on Howard Gardner, a developmental psychologist at Harvard University, proposed that dancers' interpersonal skills are the engine of artistic communication between dance partners [44].

In addition, It is also recommended to cultivate long-term affection between dance partners, such as intimacy, harmony, mutual help, mutual tolerance, mutual attraction.”

We tried our best to improve the manuscript and made some changes marked in red in revised paper which will not influence the content and framework of the paper. We appreciate for editor and eviewers’ warm work earnestly, and hope the correction will meet with approval. Once again, thank you very much for your comments and suggestions.

Round 2

Reviewer 2 Report

I appreciate for the your work earnestly.

To sum up, I recommend the manuscript for publication.

Best regards.